# NL2PROGPT: TAMING LARGE LANGUAGE MODEL FOR CONVERSATIONAL PROTEIN DESIGN

## ABSTRACT

Large Language Models (LLMs), like ChatGPT, excel in cross-modal tasks thanks to their powerful abilities in natural language comprehension, generalization, and reasoning. Meanwhile, the wealth of human-curated protein knowledge in text form presents a unique opportunity for LLMs to contribute to advanced protein design. In this work, we propose a new LLMs-based framework, namely NL2ProGPT, for macromolecular protein sequence generation that bridges the domain gap between natural and protein languages. Specifically, we first combine the protein functions and properties to create specific text guidelines for designing the protein, ensuring it follows precise controls. Second, in order to form a more informative and generalizable protein description, we explicitly inject protein structural information by clustering the embeddings from pre-trained protein language models. Third, we train a reward model to align the protein language model with the Rosetta energy function, following an RLAIF (reinforced learning from AI feedback) fashion. We empirically verify the effectiveness of NL2ProGPT from three aspects: (1) outperforms existing protein sequence design methods in different evaluations; (2) exhibits more than 90% consistency in text-to-protein generation; (3) has effective exploration potential in disordered regions.

## 1 INTRODUCTION

Recent years have witnessed remarkable progress in Natural Language Processing (NLP), driven by pre-trained Large Language Models (LLMs) Brown et al. (2020); Radford et al. (2019); OpenAI (2023) that have shown powerful abilities in natural language comprehension, generalization, and reasoning. Notably, parallels have been drawn between protein sequences and human languages, both being composed of structured elements, with proteins using amino acids as their alphabet. Protein sequences, akin to human languages, efficiently encode structure and function in their order.

Therefore, despite dissimilarities, the analogies between protein sequences and language have motivated the use of NLP in recent protein research works Lin et al. (2022); Zheng et al. (2023); Brandes et al. (2022); Nijkamp et al. (2022); Madani et al. (2020); Ferruz et al. (2022). For example, one main set of language models follows an autoregressive training strategy, where models predict successive words based on contextual information. Protein autoregressive language models, such as ProGen Madani et al. (2020), ProGen-2Nijkamp et al. (2022), RITA Hesslow et al. (2022), and Prot-GPT2 Ferruz et al. (2022), have also been investigated, highlighting the promise of autoregressive generation in the context of protein design. However, most existing methods mainly utilize protein sequential or structural information to model the intrinsic properties of protein, lacking the kind of controllable generation in a conversational way like LLMs.

Meanwhile, there exists a vast amount of human-curated knowledge in text format describing proteins' high-level properties, such as their structure domain, function, and interactions. Given the advancements in NLP's understanding and generation of human language, there's potential to apply these methods to tackle protein-related challenges, especially for conversational protein design. Simultaneously, there exist two main challenges: 1) the sparse representation of protein description in text; 2) the lack of structure constraint in LLMs' training.

To address those challenges, with respect to early attempts to use LLMs in protein generation field, we propose our model as shown in Figure 1. In this work, we propose NL2ProGPT, a generic

approach to finetune LLMs to design protein sequences of a desired field. First, we synthesize protein functions and property description texts to establish precise design guidelines, ensuring strict adherence to defined controls. Second, we enhance the informativeness and generalizability of protein descriptions by explicitly incorporating structural information, achieved through clustering the embeddings generated from pre-trained protein language models. Third, to leverage the strengths of high-level structural constraint, we employ a reward model and a Reinforcement Learning from AI Feedback (RLAIF) methodology Ziegler et al. (2019); Lee et al. (2023) to align the protein language model with Rosetta energy function Baek et al. (2021) and cluster representation score, considering generality and consistency of generated proteins.

Under textual constraints, our experimental data shows that NL2ProGPT exhibits a high degree of consistency in protein generation, with a probability of successfully satisfying textual constraints exceeding 90%. Compared with other unconstrained protein generation models, our research results show that NL2ProGPT is closer to the characteristics of natural amino acids in terms of isomeric energy analysis and self-consistent perplexity. Furthermore, by maintaining protein structural similarity, our results demonstrate that NL2ProGPT has effective exploration potential in disordered regions. In summary, NL2ProGPT demonstrates excellent performance in the field of protein generation, provides valuable insights into research in protein engineering and related fields, and is expected to promote future exploration and applications.

We summarize our contributions as follows:

- We propose our model (NL2ProGPT) that bridges the gap between protein sequence and biomedical text, achieving the goal of conversational protein design.
- We introduce to enrich the informativeness and generalizability of protein descriptions by incorporating structural information from protein representation models.
- We introduce a strategy based on RLAIF (reinforced learning from AI feedback) that finetunes our model under the constraints of structural information.
- Comprehensive experiments demonstrate the effectiveness of NL2ProGPT on text-to-protein generation, surpassing existing protein sequence design methods.

## 2 RELATED WORKS

**Large language models**: Recently, Large Language Models (LLMs) Radford et al. (2018) Radford et al. (2019) Brown et al. (2020) OpenAI (2023) with a mass of parameters have achieved remarkable success not only in Natural Language Processing (NLP) Wei et al. (2023) but also in cross-modal fields such as computer vision Yu et al. (2023), recommender systems Hou et al. (2023), biomedical text generation Luo et al. (2022), and molecule discovery Jumper et al. (2021). For instance, ChemGPT Bran et al. (2023), a GPT variant with over a billion parameters, has been introduced to understand and generate small molecules in chemistry. BioGPT Luo et al. (2022), a domain-specific generative Transformer language model, pretrained on a large corpus of biomedical literature, was evaluated across six biomedical natural language processing tasks in the article. Therefore, LLMs demonstrate strong generalization and reasoning abilities, which enable them to excel in various tasks without extensive fine-tuning, reducing computational costs. Consequently, LLMs offer an unprecedented potential to advance protein discovery, particularly in the context of text-to-protein translation.

**Protein generation models**: Protein structure design has recently witnessed significant advancements. It has evolved from traditional methods that relied on multiple sequence alignments Do et al. (2005) Thompson et al. (1994) to generate protein structures to the utilization of deep learning and statistical techniques Jumper et al. (2021) Baek et al. (2021) for more precise modeling and prediction of the three-dimensional spatial structures of proteins. Several BERT architecture-like models, such as TCR-BERT Wu et al. (2021), epiBERTope Park et al. (2022), ESM-2 Lin et al. (2022), LM-DESIGN Zheng et al. (2023), and ProteinBERT Brandes et al. (2022), have demonstrated competitiveness on the task of protein representation learning, where they are pre-trained by introducing noise to input tokens and aiming to reconstruct the original sentences. Meanwhile, these models can also be adapted for protein generation. Another category of language models relies on autoregressive training, where models predict subsequent words based on context. Protein autoregressive language models like ProGen Madani et al. (2020), ProGen-2 Nijkamp et al. (2022), RITA Hesslow

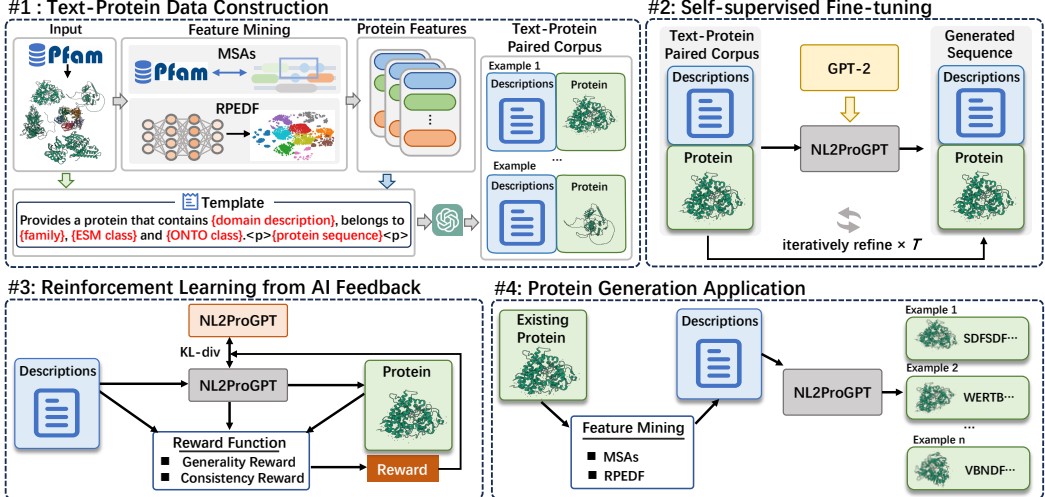

Figure 1: This diagram illustrates the NL2ProGPT workflow for protein design, which comprises four primary stages. **(1)** Protein-Text Data Construction is to construct the system prompt by constructing two different types of protein representation. **(2)** Self-supervised Fine-tuning lets Large Language Models (LLMs) perform autoregressively prediction based on the provided protein-text pairs. **(3)** Reinforced Learning from AI Feedback is executed to ensure the desired output under the structural constraints for protein design. **(4)** Protein Generation Application on producing self-consistent overall sequences for a target protein.

et al. (2022), and ProtGPT2 Ferruz et al. (2022) have also been explored, highlighting the potential of autoregressive Transformers for protein design. One similar work to our model is ProteinDT Liu et al. (2023), which also leverages textual descriptions for protein design, but adopts contrastive learning to align the two modalities.

**Protein credibility prediction**: The best way to confirm protein sequence reliability is through wet lab experiments like DMS assays, receptor binding assays, antibody tests, or thermal stability checks. However, such wet experiments require a large amount of manpower and resources, leading to the use of mathematical models for credibility prediction. For example, ProGen Madani et al. (2020) and ProGPT2 Ferruz et al. (2022) employ Rosetta Park et al. (2016) for heterogeneity energy analysis of proteins, while EvoDiff Lin et al. (2023) utilizes pLDDT and self-consistency perplexity measurements to assess the structural rationality of generated proteins, as well as secondary structure distribution to evaluate the biological properties of protein sequences. ProGPT2 and EvoDiff also possess the capability to explore disordered regions in proteins.

## 3 METHODOLOGY

### 3.1 DATA CONSTRUCTION

In this study, the data preparation phase of our approach involves randomly selecting over 1 million proteins from the Pfam dataset Bateman et al. (2004) as our training dataset. For the vocabulary representation of amino acids, we adopt the standard 25 amino acid names in IUPAC Pettit & Powell (2006). For each protein sequence, we perform two different types of feature representation construction, as shown in Figure1.

Specifically, we first use the bioinformatics tool InterProScan Jones et al. (2014) to conduct multiple sequence alignments (MSAs) of protein sequences with the Pfam database Bateman et al. (2004), which contains a large number of structural domains and other relevant information, such as protein family and conserved site, to determine the functional domains and features presented in the input sequence. This process helps capture the functional information and domain characteristics of the protein.

However, some attributes of protein in the Pfam database are quite sparse for the entire protein space (e.g., less than 150 proteins for the White spot syndrome virus structural envelope protein VP Domain) and the whole distribution appears in a long-tail form, restricting the model from generating diverse results. Therefore, secondly, we use the pre-trained protein representation model (PRM) to extract the embedded features of the protein, and then reduce protein embedded features dimensionality (RPEDF) to obtain its protein representation, thereby achieving the informativeness and generalizability enhancement of protein descriptions and further constraints on structure and function. In our research, we use ESM-2 Lin et al. (2022) and OntoProtein Zhang et al. (2022) models as examples. It is worth noting that the features extracted by ESM-2 were mainly used for protein structure prediction, making these features have certain structural representation capabilities. OntoProtein is a general framework for building protein pre-trained models using Gene Ontology structures. The features extracted by OntoProtein are often more related to Gene Ontology. The overall data processing process is as follows:

$$E_a^K = \text{PRM}(\mathbf{a}), \tag{1}$$

$$E_p^K = \text{AveragePooling}(E_a^K), \tag{2}$$

$$E_p^2 = \text{UMAP}(E_p^K), \tag{3}$$

$$C_p = \text{K-means}(E_p^2). \tag{4}$$

Specifically, we first extract the residue dimension features $E_a^K \in \mathbb{R}^{L \times K}$ of a protein $\mathbf{a}$ with residue length $L$ through the PRM. Then, we perform an $\text{AveragePooling}$ operation on the residue dimension features along the sequence dimension to obtain the overall representation feature of the protein $E_p^K \in \mathbb{R}^K$. Next, we use the UMAP algorithm McInnes et al. (2018)to reduce the dimensionality of the protein representation feature $E_p^K$ and map it to a two-dimensional space to obtain $E_p^2 \in \mathbb{R}^2$. Finally, we use the K-means clustering method to cluster the dimensionally reduced protein representation $E_p^2$, group the protein data into different clusters, and obtain the cluster representation $C_p \in \mathbb{R}$. It should be noted that the protein feature representation is first dimensionally reduced through UMAP, and then is clustered through K-means instead of clustering the protein feature representation directly through the clustering method, which can make the entire process more intuitive and reliable.

Finally, we manually construct templates and embed the obtained protein representations into the text (for example, if ESM clustering is category *1*, it is converted to the text "*ESM_1*" and embedded in the corresponding position of the template), generating descriptions for each protein. We then feed these constructed templates into ChatGPT OpenAI (2023) to obtain diverse protein text descriptions by using several prompts. These descriptions constitute the training dataset for text-protein pairs, serving as a foundation for further research and analysis.

### 3.2 SELF-SUPERVISED FINE-TUNING

Let $\mathbf{a} = (a_1, \ldots, a_{n_a})$ represent the amino acid sequence, which signifies the composition of a protein. Similarly, let $\mathbf{w} = (w_1, \ldots, a_{n_w})$ denote the protein's description. $A$ and $W$ can be defined as the input spaces for $\mathbf{a}$ and $\mathbf{w}$, respectively, such that $\mathbf{a} \in A$ and $\mathbf{w} \in W$. By merging the textual description with the amino acid sequence into the sequence $\mathbf{x} = [\mathbf{w} : \mathbf{a}]$, we create a combined sequence containing protein information and derive its probability distribution $P(\mathbf{x})$. To model this probability distribution, we employ the probabilistic chain rule and train it with the help of a neural network to minimize the negative log-likelihood on the dataset $D$:

$$P(\mathbf{x}) = \prod_{i=1}^n P(x_i | x_{<i}), \tag{5}$$

$$\mathcal{L}(D) = -\frac{1}{|D|} \sum_{k=1}^{|D|} \frac{1}{n_k} \sum_{i=1}^{n_k} \log p_\theta(x_i^k | x_{<i}^k). \tag{6}$$

This training process helps us understand the distribution of combined sequences. We have paid special attention to $P(\mathbf{a}|\mathbf{w})$, which represents the distribution of protein amino acid sequence $\mathbf{a}$ given a text description $\mathbf{w}$. To achieve this goal, we conduct an initial fine-tuning phase, as shown

in Figure 1, where we utilize the pre-trained GPT-2 model Radford et al. (2019) with text understanding capabilities as the initial state of NL2ProGPT. Subsequently, NL2ProGPT further learns conditional distributions between amino acid and protein descriptions. This process involves mapping a sequence of tokens into a vector space and processing it through multiple Transformer layers. During fine-tuning, we utilize a cross-entropy loss function to compare the model output with the true labels to provide guidance. When generating new sequences, we use the softmax function to calculate the sampled final label distribution.

### 3.3 Reinforced Learning from AI Feedback

Inspired by Ziegler et al. (2019); Lee et al. (2023), as shown in Figure 1, we consider introducing the feedback mechanism of reinforcement learning into the text-to-protein generation task.

For our text-to-protein generation task, we define the data distribution of protein text descriptions as $\mathbb{D}$, and $P$ as defined above a probabilistic strategy $P(\mathbf{a}|\mathbf{w}) = P(\mathbf{aw})/P(\mathbf{w})$: fix the text description of the protein to $\mathbf{w}$, and then use probability $P$ to generate subsequent tokens. In this paper, we denote the initial policy as is $\pi = P$, and fine-tune $\pi$ through reinforcement learning to better complete the task. The specific task is defined by the reward function $r : W \times A \rightarrow \mathbb{R}$, then we could use RL to directly optimize the expected reward:

$$\mathbb{E}_\pi[r] = \mathbb{E}_{\mathbf{w}\sim\mathbb{D}, a\sim\pi(.|\mathbf{w})}[r(\mathbf{w}, \mathbf{a})] \tag{7}$$

Our reward function $r$ mainly considers two dimensions, namely generality and consistency. In the context of generality, we investigate the conformational energies of proteins and assessed the stability and energy of various protein conformations using the Rosetta energy function Park et al. (2016), also referred to as the potential energy function. This function encompasses interactions and force fields such as van der Waals forces, charge interactions, hydrogen bonds, and virtual side-chain conformations. Generally, protein structures with lower scores are more likely to be closer to the native structure. The specific reward points are calculated as follows:

$$r_{\text{rosetta}} = \alpha - \ln(r_{\text{raw\_rosetta}} + \beta) \tag{8}$$

Among them, $\alpha$ and $\beta$ are customized bias terms, which are optimized by optimizing the original Rosetta score to better train the model.

On the consistency dimension, we considered cluster representation scores. When the generated protein matches the target protein, we award it a score of $\mu$. When there is no match, we consider the distance between the dimensionality reduction coordinates of the generated protein and the coordinates of the cluster center point. The farther the distance, the lower the reward score obtained. The specific reward points are calculated as follows:

$$r_{\text{esm}} = \begin{cases} \mu, & \text{if}(x_i^{esm}, y_i^{esm}) \rightarrow c_i^{esm} \\ \mu - \sqrt{(x_i^{esm} - x_{c_i}^{esm})^2 + (y_i^{esm} - y_{c_i}^{esm})^2}, & \text{otherwise} \end{cases} \tag{9}$$

$$r_{\text{onto}} = \begin{cases} \mu, & \text{if } (x_i^{onto}, y_i^{onto}) \rightarrow c_i^{onto} \\ \mu - \sqrt{(x_i^{onto} - x_{c_i}^{onto})^2 + (y_i^{onto} - y_{c_i}^{onto})^2}, & \text{otherwise} \end{cases} \tag{10}$$

where $\mu$ is the hit reward score of the clustering result, and $c_i^{esm/onto}$ is the cluster center point corresponding to the i-th protein.

Finally, our comprehensive award score is:

$$r = \lambda_1 r_{\text{rosetta}} + \lambda_2 r_{\text{esm}} + \lambda_3 r_{\text{onto}} \tag{11}$$

where $\lambda_1, \lambda_2$, and $\lambda_3$ are hyperparameters used to balance the contribution of each reward score.

### 3.4 Protein Generation Application

One significant challenge in protein design is the concept of inverse protein folding Hsu et al. (2022), where the objective is to select amino acid sequences that autonomously fold into a predetermined backbone structure. While some existing LLMs-based methods Madani et al. (2020); Ferruz et al.

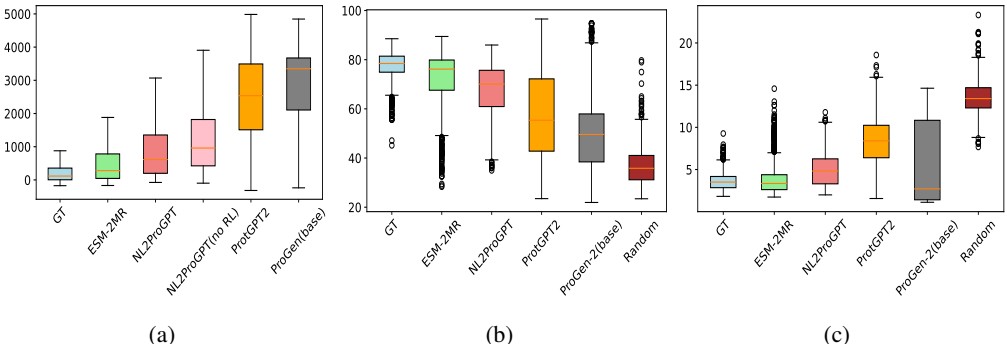

(a)                    (b)                    (c)

Figure 2: Comparison results of (a) conformational energy distributions, (b) foldability-measured sequence pLDDT distributions, (c) and self-consistency distributions. Model sizes: ESM-2MR (650M), ProGen-2 (764M), ProtGPT2 (738M), Ours (124M).

(2022); Nijkamp et al. (2022) have demonstrated success in De novo protein design, none of them enables the sequence generation given target structures due to the lack of structural constraints. In this work, we take the first step toward inverse protein folding with LLMs, where we can obtain target protein structural embedding with protein structural representation model, i.e., ESM-2 Lin et al. (2022). Then we inject the embedding's textural expression into the target protein description, making our model able to produce self-consistent overall sequences for a target protein as the results shown in Section 4.3.

## 4    RESULTS AND ANALYSIS

### 4.1    IMPLEMENTATION DETAILS

Our training dataset comprise 1,001,890 text-protein sequence pairs in total. We extend our training based on the GPT-2 architecture with the following hyperparameters: random seed (42), batch size (12), learning rate (3e-5), training epochs (20) with a warm-up step of 11,000, and the Adam optimizer. For reinforcement learning, we employ Proximal Policy Optimization (PPO) Schulman et al. (2017) with a learning rate of 1.41e-6 and a ratio threshold of 8.0. We use ESMFold Lin et al. (2022) to predict protein structures and calculate Rosetta scores based on the weight configuration of ref2015Park et al. (2016). The hyperparameters for the various reward functions, denoted as $\alpha$, $\beta$, and $\mu$, are set to 8.0, 500.0, and 1.0, respectively. The reward score weights, $\lambda_1$, $\lambda_2$, and $\lambda_3$, are assigned values of 2.0, 1.0, and 1.0, respectively.

### 4.2    GENERATION RESULTS EVALUATION

**Generality Evaluation.**  Our research focuses on assessing the quality of the generated protein sequences and examining whether the model can generate novel and structurally sound protein sequences. Therefore, we compared our NL2ProGPT (with and without reinforcement learning) with several state-of-the-art protein sequence generation methods, including ProGen-2(base) Nijkamp et al. (2022), ESM-2 Masking Reconstruction(ESM-2MR) Lin et al. (2022) and ProtGPT2 Ferruz et al. (2022). ESM-2MR is a method that employs a random 50% masking of the protein sequence, followed by sequence reconstruction utilizing the ESM-2 model. Random is obtained by randomly mutating 50% of the protein sequences.

We randomly generate 1000 protein sequences from these models and used ESMFoldLin et al. (2022) for structure prediction, followed by Rosetta scoring Park et al. (2016). The results of this study are shown in Figure 2(a), showing that the protein sequences generated by text-precise protein constraints are closer to the distribution of real data in terms of Rosetta scores than other models. In particular, it is important to emphasize that models fine-tuned with reinforcement learning perform best in this regard. Overall, our generated protein sequences may have a higher success rate when performing wet experiments.

Table 1: Comparison of consistency success rates under different text constraints.

| Method | Domain | ESM Cluster CLS | OntoProtien Cluster CLS |
|---|---|---|---|
| ESM-2MR | 0.887 | 0.791 | 0.795 |
| NL2ProGPT(no RL) | 0.980 | 0.879 | 0.902 |
| NL2ProGPT | **0.994** | **0.917** | **0.970** |

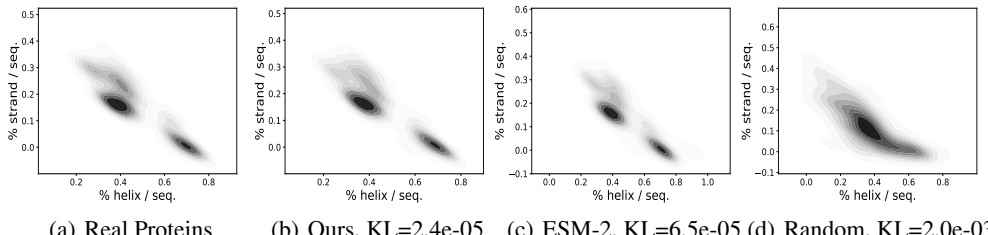

(a) Real Proteins     (b) Ours, KL=2.4e-05     (c) ESM-2, KL=6.5e-05     (d) Random, KL=2.0e-03

Figure 3: We performed an analysis of the three-state secondary structure of the generated sequences, including multivariate distributions of helical and folded structures.

As shown in Figure 2(b), we further evaluate the quality of the protein structure by calculating the average predicted local distance difference test (pLDDT) to measure the foldability of individual sequences. pLDDT not only reflects ESMFold's degree of confidence in the protein structure but also provides an assessment of the quality of the prediction. We noticed that in some cases lower pLDDT scores may be associated with the presence of intrinsically disordered regions (IDRs) in the protein (e.g. as shown in Figure 5). This phenomenon also commonly exists in many natural proteins.

In addition, we use the inverse folding algorithm ESM-IF Hsu et al. (2022) to redesign each predicted protein structure and calculate self-consistent perplexities for the originally generated sequences as shown in Figure 2(c). Lower values of self-consistent perplexity indicate that the generated structure is more consistent with the sequence, while higher self-consistent perplexity may indicate that the generated sequence is more diverse. We can observe that NL2ProGTP achieves a good balance between reliability and diversity.

**Consistency evaluation.** Since our protein generation task is based on textual constraints, it is critical to evaluate whether our model can accurately generate protein sequences that comply with the textual requirements. Considering that there is currently no model for text-to-protein generation use, we adopt ESM-2MR and Random as a strong baseline for evaluation. As shown in Table 1, compared with ESM-2MR, our model has achieved higher performance under various conditions. Interestingly, although we do not reward hitting protein domains during reinforcement learning fine-tuning, the hit rate of protein domain correlations has also been improved after fine-tuning, indicating that our model implicitly performs better in clustering.

We have also predicted the three-state secondary structure of all protein sequences using the ProtT5 model Elnaggar et al. (2020) and have calculated the KL divergence between the generated sequences and the real data secondary structure distribution. As shown in Figure 3, we observe that the protein secondary structure distribution generated by NL2ProGPT is closer to the real data than ESM-2MR. This demonstrates that NL2ProGPT is able to maintain not only high quality when generating proteins but also consistency with the natural distribution.

Additionally, we explore whether NL2ProGPT truly learns cluster representation. We randomly select 3 protein descriptions and generate 500 protein sequences for each description. Subsequently, we use ESM-2 and protT5 model to extract feature representations from all protein sequences, and calculate the Fréchet ESM-2 distance (FED) Alamdari et al. (2023) and Fréchet ProtT5 distance (FPD) Alamdari et al. (2023) respectively. Through t-SNE dimensionality reduction visualization shown in Figure 4, we can find that the sequences generated by NL2ProGPT are clearly distributed in 3 clusters as the real distribution, while the sequences generated by ESM-2MR are not clearly distinguished as our NL2ProGPT in the case of ESM-2 embedding.

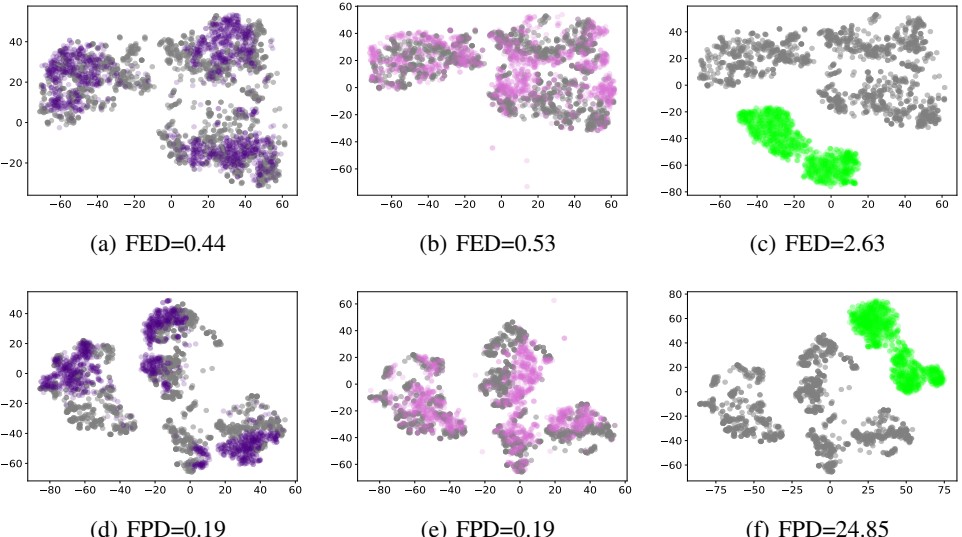

Figure 4: ESM-2 embedding (a-c) & ProtT5 embedding (d-f) distribution of generated protein sequences. Embedding for real proteins (grey), NL2ProGPT (purple), ESM-2MR (pink), and random mutations (green).

Table 2: TM-scores between protein clusters (ESM_32 and ESM_86), and comparison between real and generated clusters.

| ESM Cluster | Real | Gen. | Real V.S. Gen. |
|---|---|---|---|
| 32 V.S. 32 | 0.87 | 0.87 | 0.87 |
| 86 V.S. 86 | 0.89 | 0.88 | 0.88 |
| 32 V.S. 86 | 0.79 | 0.78 | - |

Table 3: Comparison of Conserved Site Frequency (CSF) differences between real proteins (Real) with ABC transporter-like, ATP-binding domain structural characteristics but belonging to different ESM clusters, and proteins generated by NL2ProGPT (Gen.).

| Type | ESM Cluster | CSF |
|---|---|---|
| Real | 32 | 0.23 |
| | 86 | 0.85 |
| Gen. | 32 | 0.33 |
| | 86 | 0.89 |

### 4.3 CASE STUDIES

We adopted a clustering method to represent model features, but we need to consider whether these cluster representations only reflect differences in embedded features, or whether they are biologically interpretable. Taking proteins including ABC transporter-like, ATP-binding domain, the ESM model clusters them into ESM_32 and ESM_86 respectively. We first select 500 protein samples each from the corresponding described real data and use ESMFold Lin et al. (2022) to predict the structure of each protein. Next, the TM-scores among these proteins are calculated through TMA-lign Zhang & Skolnick (2005), and the results are shown in the second column of Table 2. We can find that the structural similarity of proteins in different clusters is significantly lower than the structural similarity of proteins in the same cluster. Similarly, we use NL2ProGPT to generate 500 protein samples of each of the two cluster descriptions and also calculate the TM-score between them. The results are shown in the third column of Table 2. Compared with real data results, we can observe that NL2ProGPT indeed learns the potential structure knowledge from the ESM cluster representation, producing TM-scores highly similar to the real data's. Additionally, TM-scores computed by real and generated clusters at the fourth column of Table 2 further verify the high similarity between real and generated proteins.

At the same time, we also noticed that the ESM clustering representation also includes some other biological characteristics, such as conserved sites of proteins. Taking the proteins containing ATP-binding domain in the ESM_86 category as an example shown in Table 3, we randomly select

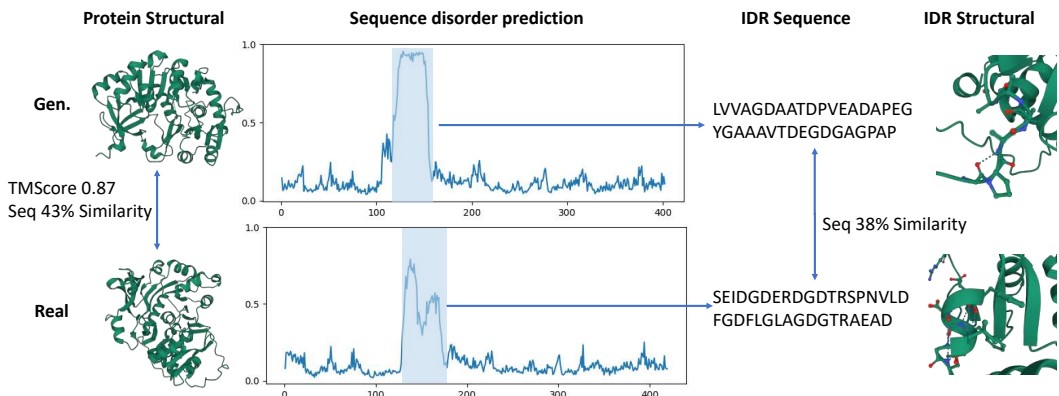

Figure 5: Representative protein disordered region analysis of NL2ProGPT and real proteins. NL2ProGPT generates predicted disorder scores and corresponding sequences for protein and true protein IDRs, and simultaneously compares sequence and structural similarities between representative rows of proteins from both.

500 real proteins and find that 85% of them had ABC transporter-like conserved sites, while only 15% in other categories. Similarly, NL2ProGPT also shows this distribution pattern, indicating that NL2ProGPT has also learned this implicit biological knowledge. Overall, our clustering representation has a certain biological meaning, and NL2ProGPT has also learned this implicit biological meaning.

In cellular functions, naturally occurring disordered regions in proteins, despite lacking a firm spatial structure, play important roles in many key biological processes, such as protein-protein interactions. Therefore, we investigate whether NL2ProGPT can explore disordered regions of proteins while meeting specific requirements. We screen proteins that have AMP-dependent synthetase/ligase domain and belong to ESM_13 and ONTO_22. As shown in Figure 5, first we use ESMFold Lin et al. (2022) to predict the three-dimensional structure of the protein sequence, and then calculate the TM-score between the real protein and the generated protein through TMAlign Zhang & Skolnick (2005). In addition, we evaluate the amino acid sequence similarity between the two through sequence alignment. Surprisingly, with only 43% amino acid sequence similarity, the TM-score is as high as 0.87. At the same time, we also use DR-BERT Nambiar et al. (2023), a tool for predicting intrinsically disordered regions of proteins. The results show that the disordered region corresponding to the protein we have generated has a higher score, while the sequence similarity of the disordered region is only 38%, and the visual difference in the structure of the disordered region is obvious. This demonstrates that NL2ProGPT can successfully explore disordered regions of proteins while maintaining protein structural similarity.

## 5 CONCLUSION

We introduce the NL2ProGPT framework, which aims to bridge the domain gap between natural language and protein language. The framework shows excellent performance and potential in multiple aspects: First, NL2ProGPT can generate macromolecular protein sequences that are close to natural proteins, indicating its potential application value in protein functional design. Secondly, the model demonstrates effectiveness in exploring disordered regions, demonstrating the ability to generate diverse protein sequences. In addition, NL2ProGPT skillfully embeds protein structural information into natural language text and shows excellent performance in natural language to protein translation consistency, emphasizing its ability to convert natural language descriptions into protein sequences. This research provides important innovations in the field of protein design. Integrating natural language and protein language opens up new research avenues for advanced protein design and provides solid support for future protein engineering and biological research.

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
