# OpenReview forum: "NL2ProGPT: Taming Large Language Model for Conversational Protein Design"
_ICLR.cc/2024/Conference — Submitted to ICLR 2024_

### Official Review · Reviewer_kJGB · 2023-10-16

**Soundness:** 2 fair
**Presentation:** 2 fair
**Contribution:** 2 fair
**Rating:** 3
**Confidence:** 5

**Summary:**

This paper trains a P(protein sequence | metadata) model, where metadata encompasses both natural language descriptions of target attributes of the protein and some control tags for structural features based on clustering structures of natural proteins. There are some interesting modeling ideas, such as fine-tuning GPT-2 and using an RL objective to reward sequences with low Rosetta energy. Samples from the model are evaluated using various sanity checks using protein structure prediction models, etc.

**Strengths:**

The paper draws on a number of modeling techniques that are popular in the modern toolbox: RLAIF, fine-tuning foundation models, using protein structure prediction tools to provide eval metrics that are cheaper than wet-lab experiments.

**Weaknesses:**

The paper's title/abstract/intro/conclusion have lots of language about the promise of a  'conversational' natural language interface for designing proteins. However, the paper does not explore such text descriptions. It just uses a simple text rendering for converting protein database entries obeying a certain schema into text. For example, "Provides a protein that contains {domain description}, belongs to {family}, {ESM class} and {ONTO class}.<p>{protein sequence}" (Fig 1). There are significant resources available for true natural language descriptions of proteins. For example, Uniprot entires have one-line name fields and also longer description fields. Further, there are lookup tables available that map GO terms, EC numbers, Pfam families, etc to free text descriptions.

Similarly, the paper seems to over-state the novelty of structured-guided design with language models. The paper says "…none of them enables the sequence generation given target structures due to the lack of structural constraints." This ignores the significant body of work using RFDiffusion+ProteinMPNN. Further, the paper's claim that it is doing structure-guided design is quite weak: they take embeddings from an ESMFold model (which presumably encode some structure information), map them down to 2 (!!) dimensions, and then cluster these. Conditioning on a cluster id is the only way that structural information is provided. The confirmation that the generated sequences have desired structures in Table 2 is quite simplistic and anecdotal.

I found the RLAIF setup quite confusing. How does it make sense to use Rosetta energy as an absolute reward function? Doesn't this need to be relative to proteins having a similar fold, similar length, etc?

The paper fine-tunes GPT-2 (which was not pretrained on protein sequences) on only 1M examples of proteins. It's unclear why this generative model was used. Why not train something from scratch, or why not train on more proteins? No ablations about the impact of using GPT-2 pretraining are provided.

**Questions:**

I found it very surprising that no recent papers from the Baker lab were cited (RFDiffusion, ProteinMPNN, etc) were cited. These are really important contributions to the field and highly related to your paper. Can you please comment on these?

I am extremely confused about why you did k-means on the 2-dimensional UMap representations. Can you provide more background about why this approach is more 'intuitive and reliable'?

I don't understand how the rosetta energy function was used as a reward, since the energy needs to somehow be normalized by the energy of ground truth proteins with the desired attributes. You say, "Generally, protein structures with lower scores are more likely to be closer to the native structure." What is 'native structure' and how is it used?

The rewards in eqs (9) and (10) have an optimum when the model just generates cluster centers, which will severely hurt diversity. When presenting your various eval metrics, I'm curious what would have happened if you had considered a simple baseline approach that just memorized a few exemplars.

I don't understand the overall evaluation setup. What does 'We randomly generate 1000 protein sequences from these models'. What metadata did you condition on? Was it 1000 different sets of metadata? How do you make this comparison fair when using models like ESM that don't have the ability to condition on metadata?

The "Protein credibility prediction" paragraph should mention that progen also confirms  wet-lab experiments.

The citation format is incorrect. It appears that there are many places where you should have been using natbib \citet{}.

---

> ### Author Response · Authors · 2023-11-22
> **Author Response to Reviewer kJGB**
>
> >Q1: I found it very surprising that no recent papers from the Baker lab were cited (RFDiffusion, ProteinMPNN, etc) were cited. These are really important contributions to the field and highly related to your paper. Can you please comment on these?
>
> A1: Our work focuses on text-to-protein generation, integrating structural information from protein representation models. (Refer to our “contributions”). RFDiffusion and ProteinMPNN are protein sequence design **given protein backbones**, which are not so-called highly related to our work.
>
> >Q2: I am extremely confused about why you did k-means on the 2-dimensional UMap representations. Can you provide more background about why this approach is more 'intuitive and reliable'?
>
> A2: Directly clustering protein representation in **high-dimensional space is unstable and hard to optimize**. The UMap could reduce the dimensionality while maintaining the spatial relationship between proteins. In addition, our experiments also confirmed that this clustering result has certain biological significance, as shown in Table 2 and Table 3.
>
>
> >Q3: I don't understand how the rosetta energy function was used as a reward, since the energy needs to somehow be normalized by the energy of ground truth proteins with the desired attributes. You say, "Generally, protein structures with lower scores are more likely to be closer to the native structure." What is 'native structure' and how is it used?
>
> A3: It is common knowledge that **lower molecular energy corresponds to higher stability**. For example, ProGen and ProtGPT2 both use Rosetta scores for evaluation (see the Related Works Section). For more details, please see (https://www.rosettacommons.org/demos/latest/tutorials/scoring/scoring). The native structure means the structure of natural proteins.
>
> >Q4: The rewards in eqs (9) and (10) have an optimum when the model just generates cluster centers, which will severely hurt diversity. When presenting your various eval metrics, I'm curious what would have happened if you had considered a simple baseline approach that just memorized a few exemplars.
>
> A4: In fact, it is not generating the cluster center point that can obtain the optimal reward. Hitting the cluster representation means hitting the dimensionally reduced region (where **the reward is the same within the region**), not a single point. Therefore, such rewards do not hurt the generation's diversity.
>
> >Q5: I don't understand the overall evaluation setup. What does 'We randomly generate 1000 protein sequences from these models'. What metadata did you condition on? Was it 1000 different sets of metadata? How do you make this comparison fair when using models like ESM that don't have the ability to condition on metadata?
>
> A5: We selected 10 text descriptions and obtained 1000 protein sequences. For ESM-2MR, we selected 10 text descriptions corresponding to protein sequences from the dataset, randomly masked 50%, and reconstructed the sequences.
>
> >Q6: The "Protein credibility prediction" paragraph should mention that progen also confirms wet-lab experiments.
>
> A6: Thanks for your advice.

---

### Official Review · Reviewer_PWDH · 2023-10-31

**Soundness:** 2 fair
**Presentation:** 2 fair
**Contribution:** 2 fair
**Rating:** 5
**Confidence:** 3

**Summary:**

This paper proposes a new LLMs-based framework “NL2ProGPT” for macromolecular protein sequence generation that bridges the domain gap between natural and protein languages. The authors train a reward model to align the protein laguage model with the Rosetta energy function, following an RLAIF fashion, and empirically verify the effectiveness of NL2ProGPT.

**Strengths:**

The authors have provided detailed explanations of their proposed methods and presented promising results.

**Weaknesses:**

The authors claim that:

“most existing methods mainly utilize protein sequential or structural information to model the intrinsic properties of protein, lacking the kind of controllable generation in a conversational way like LLMs.”

It is unclear what advantages can be brought by “generation in a conversational way.”

**Questions:**

Misc:

The citations should be enclosed by parentheses, such as using the“\citep{}” command instead of “\cite{}”.

Typo in Table 1: “OntoProtien”, “NL2ProGTP”

---

> ### Author Response · Authors · 2023-11-22
> **Author Response to Reviewer PWDH**
>
> Conversational protein generation models, by offering a more intuitive, flexible, and interactive interface, provide a powerful tool for researchers in the field of biology, potentially driving deeper breakthroughs in related research. As discussed in the introduction's LLM section, they can serve as a tool for researchers in the field of biology, facilitating discoveries and advancing related research. As stated in our paper, "Consequently, LLMs offer unprecedented potential to advance protein discovery, particularly in the context of text-to-protein translation."

---

### Official Review · Reviewer_se98 · 2023-11-01

**Soundness:** 3 good
**Presentation:** 2 fair
**Contribution:** 2 fair
**Rating:** 5
**Confidence:** 3

**Summary:**

This work studies the problem of protein design with large language models (LLMs), where the input to their model is a natural description of protein features that contain both functional and structural information via preprocessing with existing MSA tools and pre-trained protein language model (e.g., ESM2). The framework — NL2ProGPT also consists of two steps of self-supervised fine-tuning on GPT2 and reinforcement learning from AI feedback (with protein-based and cluster-based rewards) to improve the model's prediction.
They evaluate the quality of proteins generated by the proposed framework and show relatively good performance on closeness to the real-data distribution and high consistency. The authors also provide interesting findings on exploring disordered regions and case studies to understand cluster representations further.

**Strengths:**

1. The problem of protein design is important. With the rapid growth of LLMs, utilizing LLMs for protein design is a timely and interesting problem.
2. The design of the NL2ProGPT framework seems to be novel in terms of integrating existing techniques used for LLMs with natural languages and techniques specified for protein learning.
3. As measuring the generation of protein is still open research, the paper makes a good effort in quality evaluation and shows a good performance of the proposed method.
4. I appreciate the effort of the authors in providing the case study

**Weaknesses:**

- Correctness/Soundness of the framework:
    - Evaluation: the method seems to use the same model in the framework for evaluation. For instance, they use ESM2 to embed the structure with reward constraints to an ESM-based cluster and use ESMFold (built on ESM2) for structure prediction evaluation. Also, they use the consistency (with Rosetta) reward in Step 3 to constrain the model and evaluation. This may raise the question of model performance benefits from the inductive bias of these pretrained models and tools.
    - The paper claims to embed the structural information into the description, but it’s doubtful how much the structure is preserved. First, though the ESM2 paper claims their embeddings have structural information, it is still implicit. Second, though the case study shows some insight into the cluster representation, it’s unclear how much information UMAP (into 2-D) and k-mean can preserve, as we know the loss of information after the dimension reduction and the difficulty of clustering.
- Results:
    - In Figure 2, it doesn’t seem the proposed method achieves the best performance in any measure. For instance, a similar approach — ESM2-MR model is closer to the GT and performs better in the first one.
    - As NL2ProGPT is not the first approach combining natural language and protein (e.g., proteinDT), this raises the question of motivation in which scenario the proposed method is necessary.
- Novelty/Originality: while I appreciate the novelty in integration methods for protein design, each framework component seems to be incremental in the design for protein learning.
- Writing or Presentation: Overall, the paper is easy to follow, but the presentation is not at the quality of the top conference and should be improved.
    - Typos: there are a few typos, such as missing space right before the citation on page 1 (ProGEn-2Nij), (ref2015Park), page 6 (-2(base)), …. These typos somewhat indicate that the paper was not properly proofread.
    - I can not find the appendix or detailed description of the model, settings, and template. It should be more useful for understanding to provide the sample template.
    - (Optional) The writing should be improved to be more concise. Some sentences and claims are  vague and less precise, e.g., “This training process helps us understand the distribution of combined sequences.”  or “Overall, our generated protein sequences may have a higher success rate when performing wet experiments.” Furthermore, the notations, e.g. aw can also be improved for consistency.
    - I didn’t find the description/definition of the TM score in the paper.
    - Minor: For Figure 1, step 1, the figure of ChatGPT seems to be a cropped version of the ChatGPT official icon without modification.

**Questions:**

Together with previous questions, I have some clarification questions:

1. For structure embedding, have you considered other methods, such as explicitly embedding structure from AlphaFold generation, which some recent papers use?
2. For step 1, how do you improve the diversity with ChatGPT? Do you also input the protein sequence to ChatGPT?
3. For step 2, what are the input and output? From the figure, it seems like a pretraining step with self-supervision (next-token prediction). Still, the description in the paper says it’s p(a|w), meaning predicting amino acids from the description. Can you elaborate on this step?
4. For step 2, what is the initial state of GPT2? Which checkpoint is that?
5. For step 4, how do you control the diversity of generated sequences given an input protein?
6. How well do they cluster in 2-d of UMAP?
7. Terminology: why do you call it conversational protein design? It may be confusing to the dialog or conversation-based LLM.

---

> ### Author Response · Authors · 2023-11-22
> **Author Response to Reviewer se98**
>
> >Q1: The method seems to use the same model in the framework for evaluation. For instance, they use ESM2 to embed the structure with reward constraints to an ESM-based cluster and use ESMFold (built on ESM2) for structure prediction evaluation. Also, they use the consistency (with Rosetta) reward in Step 3 to constrain the model and evaluation. This may raise the question of model performance benefits from the inductive bias of these pretrained models and tools.
>
> A1: We sincerely do not agree that our evaluation is unfair because ESMFold is only used for structure prediction, and the real score is obtained by the Rosetta scoring function. In addition, we do not think that using RL is unfair for comparison. First, we use multiple evaluation metrics to verify our model’s superior performance than other methods as shown in Figure 2. Second, we also report the performance of our method without RL in Table 1, still showing a better performance than the baseline model.
>
> >Q2: The paper claims to embed the structural information into the description, but it’s doubtful how much the structure is preserved. First, though the ESM2 paper claims their embeddings have structural information, it is still implicit. Second, though the case study shows some insight into the cluster representation, it’s unclear how much information UMAP (into 2-D) and k-mean can preserve, as we know the loss of information after the dimension reduction and the difficulty of clustering.
>
> A2: From Table 2, we can see that our method can preserve quite high structural information in the generated proteins. It also should be emphasized that we provide a new **flexible** way to incorporate structural information for the protein language model that has not been explored before.
>
> >Q3: In Figure 2, it doesn’t seem the proposed method achieves the best performance in any measure. For instance, a similar approach — the GT model performs better in the first one.
>
> A3: It is important to clarify that in Figure 2, **'GT' represents the ground truth protein**.
>
> >Q4: As NL2ProGPT is not the first approach combining natural language and protein (e.g., GTprotein), this raises the question of motivation in which scenario the proposed method is necessary.
>
> A4: Could you give a reference of GTprotein or do you mean the ground truth model ?
>
> > Q6： Questions:
> 1. Our approach exhibits strong scalability, allowing for the incorporation of additional protein representation models in the future.
>
> 2. We exclusively leverage ChatGPT to enrich the textual description of proteins. Only text is inputted; protein sequences are not included.
>
> 3. Our methodology involves concatenating protein textual descriptions with protein sequences for unified input during self-supervised training. Our focus is solely on obtaining protein sequences through text. During inference, protein sequences are fully derived from the model through textual descriptions.
>
> 4. We utilize the official GPT-2 checkpoints provided by Hugging Face: https://huggingface.co/gpt2.
>
> 5. GPT employs sampling during generation, providing control over diversity by adjusting the sampling strategy.
>
> 6. The combination of Figure 4, Table 2, and Table 3 indirectly validates the effectiveness of our clustering approach.
>
> 7. Since our model is developed as a protein generation model within a conversational system, we refer to it as conversational protein design.

---

> ### Comment · Reviewer_se98 · 2023-11-22
> **Re: clarification for Q3 & Q4**
>
> Hi,
>
> Thanks for your response.
>
> * Regarding Q3, I meant the 'ESM-2MR' model  -- is closer performance to the GT (I'm sorry for the typo).
> * Regarding Q4, the references for using text with protein design are the ProteinDT model (A Text-guided Protein Design Framework, Liu et. al. 2023. https://arxiv.org/pdf/2302.04611.pdf), or ProGen family (Large language models generate functional protein sequences across diverse families, Madani et. al., 2023, https://www.nature.com/articles/s41587-022-01618-2).

---

> > ### Author Response · Authors · 2023-11-22
> > **Re: Re: clarification for Q3 & Q4**
> >
> > Thanks for your clarification, and we will give the new explanation as follows:
> >
> > For Q3, ESM-2MR indeed performs better than our method in Figure 2(a), which is not very surprising. We take ESM-2MR as a strong baseline because, instead of de novo protein generation, ESM-2MR employs a random 50% masking of the protein sequence, followed by sequence reconstruction utilizing the ESM-2 model. Therefore, the results of ESM-2MR can be more similar to GT proteins in certain metrics.
> >
> > For Q4, we agree that our model is not the first approach combining natural language and protein. However, to the best of our knowledge, we are the first work to leverage LLMs and protein representation models for conversational protein design, unlike ProteinDT (not open-sourced) and ProGen family that train the model from scratch and omit the generative prior knowledge in LLMs or protein representation models.

---

### Official Review · Reviewer_Gu8r · 2023-11-03

**Soundness:** 2 fair
**Presentation:** 2 fair
**Contribution:** 2 fair
**Rating:** 5
**Confidence:** 2

**Summary:**

The paper proposes to train a joint model on both the protein and text modalities (optionally with RL with some rewards around generality and consistency). The model is then used to generate proteins, sometimes with textual constraints that are key to the authors' approach.

The models are evaluated with respect to several related works on both the generality and consistency dimensions.

**Strengths:**

I think the idea of building joint protein text representations for controllable protein generation is important and has a lot of promise.

**Weaknesses:**

-Overall I have some questions about the paper below that I feel are critical to my understanding both to understand the method and make sure the evaluation is fair.

(1) I don't quite understand how ChatGPT is used to generate the text descriptions. It says something like:

"We then feedthese constructed templates into ChatGPT OpenAI (2023) to obtain diverse protein text descriptions
by using several prompts. These descriptions constitute the training dataset for text-protein pairs,
serving as a foundation for further research and analysis."

In Section 4.1 it also says:
"Our training dataset comprise 1,001,890 text-protein sequence pairs in total."

Are these training examples from the above process with ChatGPT? If so, how did you do any verification on the quality of this dataset?

(2) Given that some of them models use text as inputs like the authors' approach and some do not e.g. Progen I am a bit confused as to how all the models are compared e.g. is each model fed a different input and what are these inputs?

(3) When evaluating for generality and consistency are the metrics used the same as that were used for RL? (in which case it would be unfair since the model would be overfitting on the reward). Some clarification would be great.

**Questions:**

See questions above.

---

> ### Author Response · Authors · 2023-11-22
> **Author Response to Reviewer Gu8r**
>
> > Q1： I don't quite understand how ChatGPT is used to generate the text descriptions. Are these training examples from the above process with ChatGPT? If so, how did you do any verification on the quality of this dataset?
>
> A1： We enriched textual descriptions using ChatGPT, specifically by leveraging the generated text from templates through ChatGPT to diversify the textual descriptions. For instance, the original statement "Provides a protein that contains GNAT domain and belongs to ESM_51 and ONTO_38." was transformed into "This protein presents attributes encompassing the GNAT domain and falls under the classifications of ESM_51 and ONTO_38." This process focuses solely on **enhancing the diversity of textual descriptions**, ensuring that the quality of the dataset remains unaffected.
>
> >Q2: Given that some of them models use text as inputs like the authors' approach and some do not e.g. Progen I am a bit confused as to how all the models are compared e.g. is each model fed a different input and what are these inputs?
>
> A2: In fact, we mainly compare the **quality** of generated protein sequences through three metrics, i.e., conformational energy distributions, foldability-measured sequence pLDDT distributions and self-consistency distributions. Therefore, for those methods without text inputs, the protein sequences are generated without text constraints.
>
> >Q3:  When evaluating for generality and consistency are the metrics used the same as that were used for RL? (in which case it would be unfair since the model would be overfitting on the reward). Some clarification would be great.
>
> A3: We sincerely do not think that using RL is unfair for comparison. First, we do not find any overfitting evidence in our experiment results. Second, we use multiple evaluation metrics to verify our model’s superior performance than other methods as shown in Figure 2. Third, we also report the performance of our method without RL in Table 1, still showing the better performance than the baseline model.

---

### Meta-Review · Area_Chair_Hb4j · 2023-12-19

**Metareview:**

The paper proposes a training approach to learn text to protein generation model.

Strength:
The paper construct a text-protein dataset and use ChatGPT to create diverse description of text. The paper also uses RL method based on Rosseta energy reward.

Weakness:
Some technical part of the method is not described clearly (e.g. structure based design, RL reward).
The evaluation and metrics are not convincing. plDDT only tells whether a protein could reasonably fold.

Regarding reviewer's comment:
1. Reviewer Gu8r's first comment on weakness is already addressed by the author.
2. Reviewer PWDH's review is too short. AC has disregarded this review.
3. Reviewer kJGB's first comment on weakness (comparing to RFDiffusion, ProteinMPNN) is already addressed by the author. this paper is for text to protein generation, rather than protein generation based on backbone. They are distinct tasks.

**Justification For Why Not Higher Score:**

The paper still has major issues on clarity and evaluation.

**Justification For Why Not Lower Score:**

N/A

---

### Decision · Program_Chairs · 2024-01-16

Reject